# Nutritional Characterization of Annual and Perennial Glassworts from the Apulia Region (Italy)

**DOI:** 10.3390/foods14193433

**Published:** 2025-10-07

**Authors:** Luigi Giuseppe Duri, Lucia Botticella, Corrado Lazzizera, Enrico Vito Perrino, Angelica Giancaspro, Anna Rita Bernadette Cammerino, Anna Bonasia, Antonio Elia, Giulia Conversa

**Affiliations:** Department of Agriculture, Food, Natural Resources and Engineering, University of Foggia, Via Napoli 25, 71100 Foggia, Italy; luigi.duri@unifg.it (L.G.D.); lucia.botticella@unifg.it (L.B.); corrado.lazzizera@unifg.it (C.L.); enrico.perrino@unifg.it (E.V.P.); angelica.giancaspro@unifg.it (A.G.); annarita.cammerino@unifg.it (A.R.B.C.); anna.bonasia@unifg.it (A.B.); antonio.elia@unifg.it (A.E.)

**Keywords:** Halophytes, mineral nutrients, antioxidants, phenols, iodine, *Salicornia europaea*, *Sarcocornia fruticosa*, *Arthrocaulon macrostachyum*

## Abstract

Halophytes are increasingly recognized as sustainable crops that offer a wide range of nutrients. This study provides a nutritional characterization of annual (*Salicornia europaea*) and perennial (*Sarcocornia fruticosa*, *Arthrocaulon macrostachyum)* species of glasswort, collected from different coastal habitats in southern Italy. *S. europaea* was also cultivated under non-saline conditions. Results showed differences in mineral content, and bioactive compounds among genotypes, but they were modulated by environmental conditions, leading to significant site-specific variation. *S. europaea*, regardless of the collecting sites, exhibited the highest concentration of minerals (K, Ca, and Mg), chlorophylls, carotenoids, and phenolic compounds as well as antioxidant activity. *A. macrostachyum* stood out for its high flavonoid and sterol content, exhibiting other nutritional traits comparable to *S. europaea* when collected in a more arid site. *A. macrostachyum* and *S. fruticosa* displayed similar compositional features, showing the highest anthocyanin and iodine (187.8 µg 100 g^−1^ FW, on average) content. Sodium and potassium—critical for hypertension management—varied, exceeding the recommended Na/K ratio (1) for human consumption, especially in *A. macrostachyum* grown close to the sea. The most promising result was observed in non-saline *S. europaea* and in an *A. macrostachyum* sample (1.7, on average). Overall findings confirm the potential of both annual and perennial glassworts as nutritionally rich, sustainable crops for marginal environments.

## 1. Introduction

Soil salinization and freshwater scarcity are growing global issues, making the increasing salinization of arable land a critical point of land degradation [1]. This, coupled with the increasing challenges of climate change, is a goal that the Food and Agriculture Organization of the United Nations [2] considers a priority, highlighting the importance of research on sustainable and resilient food sources. Halophytes are a group of plants that thrive in saline environments (coastal and inland areas) and have attracted significant interest as a potentially sustainable option for agriculture [3]. These plants can grow in marginal lands that face multiple constraints such as drought, fluctuating temperatures, high light irradiance, and elevated salinity [4,5], without competing for primary resources like freshwater. Many halophyte species are edible plants possessing unique biochemical properties that make them valuable as novel foods [6]. To complete their life cycle in saline environments, these plants have developed adaptive mechanisms that involve the production of secondary antioxidant metabolites [4,5].

Salicornioideae (Amaranthaceae/Chenopodiaceae *sensu* The Angiosperm Phylogeny Group [7]) is an important halophyte subfamily with many genera, such as Salicornia L., *Sarcocornia* A.J. Scott. and *Arthrocaulon* Piirainen & G. Kadereit.

Among these genera, *Salicornia europaea* Auct. (syn.: *Salicornia ramosissima* Auct., *Salicornia patula* Duval-Jouve, *Salicornia perennans* Willd. subsp. *perennans*) [8,9] is an annual taxon with a Western European geographic distribution colonising periodically inundated muds and sands of marine or interior salt marshes.

*Sarcocornia fruticosa* (L.) A.J. Scott (homotypic synonyms: *Arthrocnemum fruticosum* (L.) Moq., *Salicornia europaea* var. *fruticosa* L., *Salicornia fruticosa* (L.) A.J. Scott) [8,9,10] is a perennial species with a Euro-Mediterranean and South African distribution that colonizes coastal salty marshes periodically flooded by tides with close to marine salinity. It grows in silty-clayey soils, poorly drained, flooded for more or less long periods by salt water.

*Arthrocaulon macrostachyum* (Moric.) Piirainen & G. Kadereit homotypic synonyms: *Arthrocnemum fruticosum* var. *macrostachyum* (Moric.) Moq., *Arthrocnemum macrostachyum* (Moric.) K. Koch, *Salicornia fruticosa* var. *pachystachya* W.D.J. Koch, *Salicornia macrostachya* Moric. [8,9,10] is a Macaronesian Mediterranean distribution taxon which differs ecologically from *S. fruticosa* as it prefers salty coastal places with salinity higher than marine and often with a rocky substrate. It is in salt marshes, in sites not directly affected by tide variations, where they grow on the reliefs between the ponds.

Due to difficult taxonomy, leafless, succulent and fleshy stems of species belonging to *Salicornia* [11,12] and, more recently, those of *Sarcocornia* [13] and *Arthrocaulon* [14] genera are commercialised and/or consumed mixed as a green leafy vegetable, commonly known as glassworts or, particularly in most European countries, as ‘sea asparagus’ [13].

The product is considered a niche offering because this salt-tolerant vegetable has traditionally been harvested from wild plants and sold in local markets. However, the expanding cultivation of this vegetable in both soil [15] and soilless systems [16] has been driven by increasing demand. As a result, its commercial economic importance is on the rise [17].

Halophytes have garnered increasing attention not only for their resilience to salinity but also for their potential as nutrient-rich, sustainable food sources. Beyond accumulating common mineral nutrients like sodium and potassium, these plants are recognized for their ability to concentrate essential trace minerals. Among these, iodine is of particular significance; it is a vital micronutrient for human health, with deficiencies remaining a pressing global health issue (involved in the synthesis of thyroid hormones, which regulate various physiological processes [18]). However, a complete nutritional evaluation requires a risk-benefit analysis, which must also consider the presence of antinutritional factors such as oxalate, which can inhibit mineral absorption [19], and an elevated sodium-to-potassium (Na/K) ratio, a factor linked to hypertension risk [20]. In the Mediterranean basin, studies evaluating various aspects of the nutritional composition of these plants have been conducted mainly in the Iberian Peninsula and some North African and Middle Eastern countries. The genus *Salicornia* is rich in compounds associated with oxidative stress (drought, salinity), such as vitamin A, fatty acids, ascorbic acid, carotenoids and phenolic compounds, as discussed in the reviews by Cárdenas-Pérez et al. [12], Ekanayake et al. [17], and Sánchez-Gavilán et al. [21]. Similar nutritional aspects were noted in species *Sarcocornia* [22] and *Arthrocaulon* [14,23]. However, considering the effects of both genotype and environmental factors, along with their interactions, there have been reports of changes in the nutritional profile of glasswort products. Higher antioxidant and mineral contents were observed in the wild *Sarcocornia perennis* than in *Salicornia ramosissima* collected in the same salt marshes [22]. Changes in fatty acid and phenolic profiles were detected between *Arthrocnemum* and *Sarcocornia* genera, despite collecting from the same areas in Spain and Portugal [24]. Cultivated *Salicornia* ecotypes exhibited enhanced fatty acid (total and profile) compared to those of the *Sarcocornia* [25].

Besides the genotype, the pedoclimatic conditions present in natural habitats and/or cultivation systems significantly influence the nutritional composition of glassworts.

Wild *Salicornia patula* had a variable phenol profile based on its mainland or coastal origin [26]. Similarly, the fatty acid profiles of *S. ramosissima* differed depending on their geographical origin [27]. When ecotypes of *Sarcocornia* and *Salicornia* were cultivated in a soilless system with increasing seawater salinity, they showed increased mineral, carotenoid [28], and phenolic content [25]. Drought stress in cultivated *S. ramosissima* also led to an increase in total phenols, fatty acids, and oxalates [29]. Overall, wild *Sarcocornia* had higher levels of carbohydrates, lipids, and phenolic compounds [30] but lower mineral content [31] compared to cultivated varieties.

In the Apulian region (Italy), both annual and perennial glassworts are consumed, mainly in the Foggia province, characterised by wide coastal salt marshes, followed by the Brindisi and Taranto areas, through the gathering of wild plants. In Foggia province, the cultivation of the annual *Salicornia* species, which started more than 40 years ago for family consumption, has now evolved into a cash crop covering a few dozen hectares (Conversa, personal communication). Despite their importance for food purposes, to our knowledge, there is a lack of comprehensive research characterizing the nutritional profiles of both cultivated and wild Apulian glassworts. Based on the above, this study aims to provide a detailed characterization of mineral and metabolite profiles (antioxidative compounds and antinutritional traits) of *Salicornia*, *Sarcocornia* and *Arthrocaulon* genera cultivated or collected along the coasts of the Apulia region. While this study primarily provides a detailed biochemical characterization of cultivated and wild Apulian glassworts, the dietary implications of select analytes of significant public health concern—notably sodium, iodine, and the dietary Na/K ratio—will be discussed within the context of our findings.

## 2. Materials and Methods

### 2.1. Sampling Area and Glasswort Description

Glasswort samples were collected in Apulian coastal areas (the geographical coordinates are in the WGS84 system):(1)Margherita di Savoia Saltworks (MSS), [Margherita di Savoia Barletta–Andria-Trani province (BT)] (site a: 41°25′54.39″ N; 16°0′31.395″ E; sito b: 41°25′39.512″ N; 16°0′28.46″ E);(2)Laguna del Re Oasis (LRO) [Manfredonia, Foggia province (FG)] (site a: 41°34′40.8″ N; 15°53′5.8″ E; site b: 41°34′47.86″ N; 15°53′0.12″ E);(3)Varano lake (VL) (Cagnano Varano, FG) (41°54′51.5″ N; 15°48′17.6″ E);(4)Monti d’Arena-Bosco Caggione (MA-BC) [Taranto province (TA)] (40°20′54.4″ N; 17°22′31.9″ E);(5)‘Spirito Contadino’ (SC) commercial farm (Borgo Tressanti, FG) (41°23′44.8″ N; 15°49′25.2″ E).

Samples of wild perennial species were collected at (1) MSS site a (*Sarcocornia fruticosa* L.), and site b (*Arthrocaulon macrostachyum* Moric.), (2) VL (*Sarcocornia fruticosa* L.), and (3) MA-BC (*Arthrocaulon macrostachyum* Moric.). At location LRO, we harvested samples of glasswort both from the wild perennial (*Arthrocaulon macrostachyum* Moric., site a) and cultivated annual plants (*Salicornia europaea* Auct. site b); at site SC, conventionally cultivated (no saline conditions) annual plants (*Salicornia europaea* Auct.) were collected.

Locations MSS, LRO, and VL were within coastal marsh wetlands with different soils, from predominantly sandy to clayey, to rarely rocky, whereas MA-BC is a fully rocky coastal site. The SC site falls within the Tavoliere plain, where intensive non-saline agriculture is practised [32].

In the MSS area, *S. fruticosa* (*S. fruticosa*-MSS) plants were located along the edge and at the mouth of the brackish water Aloisa canal (site a), whereas *A. macrostachyum* (*A. macrostachyum*-MSS) was found adjacent to the salt pan area, 0.5 km away from the coast (site b). *A. macrostachyum* was also gathered at the MA-BC (*A. macrostachyum*-MA-BC) site, on the rocky seacoast, 1–2 m high. At the VL site, *S. fruticosa* (*S. fruticosa*-VL) was in a brackish area that is rarely flooded, close to the mouth of the salt Varano lake. The LRO location is characterized by the alternation of flooded areas and dry land for agricultural use, both with highly saline and sodic, often clayey soil [33]. Specifically, *A. macrostachyum* (*A. macrostachyum*-LRO) was along the edges of a brackish water stream (site a), whereas *S. europaea* was cultivated on reclaimed land for agriculture (site b) (*S. europaea*-LRO). The samples collected were summarized in Table 1, and Figure 1 shows the location of the sampling sites.

For cultivation, plantlets of *S. europaea* were provided by the nursery ‘Aurora’ (San Nicandro Garganico, FG—Italy). The transplant was performed in April (SC) and May (LRO) 2023. At site LRO-b, the soil and irrigation water’s electric conductivity (EC) were 1.9 and 21.5 mS cm^−1^, respectively [33]. In contrast, EC of soil and irrigation water was, respectively, 0.2 and 0.6 mS cm^−1^ at the SC site.

Edible tips (2 kg) from at least 20 plants of each site were randomly collected during the April–June 2023 period (wild samples) and the August-September period (cultivated samples). For each site, the harvested plants were divided into three subsamples, then washed with distilled water, dried with paper, and subjected to freeze-drying (ScanVac CoolSafe 55-9 Pro; LaboGene ApS, Lynge, Denmark) for dry mass determination (g·kg^−1^ of fresh weight, FW) and subsequent analyses, performed with six repetitions for each subsample (n = 18).

### 2.2. Inorganic Ions and Iodine Determination

#### 2.2.1. Anions and Cations

Anion concentrations were determined by ion chromatography (Dionex ICS 3000; Dionex ThermoFisher Scientific, Waltham, MA, USA) after extraction from dried samples (0.5 g) with 50 mL of eluent solution (3.5 mM sodium-carbonate and 1.0 mM sodium bicarbonate) following the procedure described by Bonasia et al. [34].

Cation concentrations were analysed by ion chromatography (Dionex ICS 3000), equipped with isocratic pump, conductivity detector, auto-sampler (AS-DV), suppressor (DRS-600; 4 mm), IonPack CS12A column (ThermoFisher Scientific, Waltham, MA, USA), after extraction from dried samples (0.3 g), previously ashed and acid digested, as detailed in Conversa et al. [35].

Anions and cations were identified by their retention times, compared with standards, using Dionex Chromeleon software (version 6.80, Thermo Scientific, Waltham, MA, USA) for peak area analysis. The results are expressed as g·kg^−1^ FW.

#### 2.2.2. Iodine

Iodine determination was performed using a spectrophotometer (Shimadzu UV-1800, Shimadzu Italia S.R.L., Milan, Italy) set at 454 nm. The extraction method was detailed in Somma et al. [36]. The quantification was determined using a calibration curve (0–12 μg L^−1^; R^2^ = 0.996). The results are expressed as μg 100 g^−1^ FW.

### 2.3. Pigments and Antioxidant Compounds

#### 2.3.1. Chlorophylls, Carotenoids and Anthocyanins

Chlorophyll *a*, *b*, and total carotenoid concentrations were determined by modifying the spectrophotometric method proposed by Sumanta et al. [37] and detailed in Bonasia et al. [38]. Chlorophyll results are expressed in µg g^−1^ FW, while the carotenoids were in mg 100 g^−1^ FW.

For anthocyanin determination, the extract method was reported by Sims and Gamon [39] and detailed in Bonasia et al. [38]. Total anthocyanin content was calculated as cyanidn-3-glucoside (c.g.), using the corrected absorbance and a molar absorbance coefficient for anthocyanin at 525 nm of 26,900 L mol^−1^ cm^−1^ [40]. Anthocyanin results are expressed as mg c.g. 100 g^−1^ FW.

#### 2.3.2. Phenols and Flavonoids

Total phenols and flavonoids extraction were carried out on 30 mg of the sample with 1 mL of water:methanol (20:80, *v*:*v*) at room temperature in an ultrasonic cleaner bath for 15 min; then the mixture was centrifuged in a refrigerated centrifuge (14,000 rpm, 15 min, 4 °C) and the supernatant was collected. The extraction was repeated two times, and the supernatants were combined. The extracts were stored at −20 °C and measured within 24 h.

Total phenolic concentration was determined according to the method of Singleton and Rossi [41]. Flavonoid content was determined using the AlCl_3_ method [42]. The detailed procedures were described in Bonasia et al. [38]. The absorbances were read at 750 and 415 nm, respectively, for phenols and flavonoids (Shimatzu UV-1800, Shimadzu Scientific Instruments, North America, Columbia, MD, USA). The results are expressed in mg 100 g^−1^ of FW as gallic acid equivalents (g.a.e.) for phenols and in quercetin equivalent (q.e.) for flavonoids.

#### 2.3.3. Phytosterols

Total phytosterols were quantified as reported by Xiang et al. [43] with some modifications. In a screw-cap tube, a lyophilized glasswort sample (100 mg) was mixed with 3 mL of sulfuric acid solution (0.5 mol L^−1^ in 95% ethanol) and vortexed for 10 s. The addition of tert-butyl-hydroquinone (3 mg) prevented the thermal decomposition of phytosterols. Acid hydrolysis was performed at 55 °C for 1 h. Alkali saponification (55 °C for 1 h) was carried out by adding 3 mL of sodium hydroxide solution (2.0 mol L^−1^ in 95% ethanol). Then, 3 mL of sodium chloride solution (1.0 mol L^−1^) was added to prevent emulsification and facilitate hexane extraction. A double extraction was performed by adding hexane (5 mL). After stirring the mixture (15 min), the two supernatants were combined. Subsequently dried using a rotary evaporator (Strike 300, Steroglass, San Martino in Campania, AV, Italy) and then dissolved in absolute ethanol (1.0 mL), creating the crude extract.

The crude extract was purified by SPE column with a 500 mg C18-cartridge (Strata C18-E, Phenomenex, Castel Maggiore, BO, Italy). The complete SPE procedure includes methanol activation, water equilibrium, sample loading, and elution.

The concentration of total phytosterols was measured according to the spectrophotometry method described by Feng et al. [44], placing 10 g of FeCl_3_ (6H_2_O) into a volumetric flask (100 mL) and diluting with phosphoric acid (85%) to reach the volume. Then, 1.5 mL of FeCl_3_ solution (10%), transferred to a volumetric flask (100 mL), was diluted with concentrated sulfuric acid. The sulfuric-phosphoric-FeCl_3_ reagent (2 mL) was combined with stigmasterol in ethanol solutions (4 mL) at concentrations ranging from 19 to 45.5 μg mL^−1^ and with the sample extract solution (4 mL). Left reacting for 15 min, afterwards the absorption was measured using a spectrophotometer at a wavelength of 520 nm. The concentration was calculated based on a standard curve. The results were expressed as mg of stigmasterol equivalent per 100 g^−1^ FW.

### 2.4. Antioxidant Capacity

We assessed the antioxidant capacity by employing three assays to determine the radical scavenging capacity (differentiating the lipophilic fraction from the hydrophilic one): ABTS (2,2′-azinobis-3-ethylbenzothiazoline-6-sulfonic acid), DPPH (2,2-diphenyl-1-picrylhydrazyl), and FRAP (Ferric Reducing Antioxidant Power) methods [45,46], described in Conversa et al. [47]. The detection of three assays was expressed in µmol T.E. g^−1^ FW, where T.E. corresponds to Trolox (6-hydroxy-2,5,7,8-tetramethylchroman-2-carboxylic acid) equivalents. For DPPH and FRAP assay, the Trolox stock solution was 1000 µM and the concentration range of the calibration curve was 0–800 µM; for ABTS test, the Trolox stock solution was 1100 µM, and the concentration range of the calibration curve was 0–15 µM.

### 2.5. Statistical Analysis

One-way statistical analysis was performed with the Statistical Analysis System software (SAS 9.1; SAS Institute, Cary, NC, USA) using the General Linear Model (GLM Proc of the SAS 9.1 Software). The mean comparison was performed using Tukey’s HSD test with a significance level of α = 0.05. The correlation between the phenolic and flavonoid compounds and the antioxidant activity of glassworts was evaluated using Spearman’s rank correlation coefficient.

The principal component analysis (PCA) was performed using the PAST3 Software (http://folk.uio.no/ohammer/past, accessed on 22 July 2025) on mean standardized data. The data standardization was performed using the formula: (x-mean)/standard deviation. The data matrix considered all glasswort collected samples with relative replications.

## 3. Results

### 3.1. Dry Mass and Mineral Concentrations

The examined glassworts had similar dry mass (DM) concentrations, except for the highest value for *Arthrocaulon macrostachyum* samples collected in Margherita di Savoia saltworks (MSS) (*A. macrostachyum*-MSS) and the lowest one registered for *A. macrostachyum* sample from Laguna del Re Oasis (*A. macrostachyum*-LRO) (Table 2).

Fluctuation in sodium concentration ranged from 6.9 to 20.9 g·kg^−1^ FW, with *A. macrostachyum* samples exhibiting a larger variability (Table 2). The highest K, Mg and Ca levels were found for *S. europaea*-SC and -LRO, especially compared to *A. macrostachyum*-LRO samples.

*A. macrostachyum*-MSS showed a higher chloride concentration, whereas Cl ranged from 8.9 to 21.7 g·kg^−1^ FW in the other sampled plants. Reduced Cl levels were registered for *Sarcocornia fruticosa* from MSS (*S. fruticosa*-MSS) (Table 2).

Nitrate level was the highest in *A. macrostachyum* collected from Taranto province locality (*A. macrostachyum*-MA-BC) and *Salicornia europaea* cultivated at the commercial farm (*S. europaea*-SC). This anion decreased in other genotypes up to not detectable values in *A. macrostachyum*-LRO (Table 2).

Variability in oxalate levels was observed, with *Sarcocornia fruticosa* from Varano Lake (*S. fruticosa*-VL) showing the highest value, followed by samples belonging to *A. macrostachyum* species.

The maximum level of iodine was observed for genotypes collected from the LRO site (*S. europaea*-LRO and *A. macrostachyum*-LRO) while the lower values were found in samples collected from MSS (both *S. fruticosa* and *A. macrostachyum*) as well as in *S. europaea* from the SC farm (Table 2).

The Na/K ratio was higher in *A. macrostachyum*-MA-BC sample (5.2), and showed a difference only compared to the *A. macrostachyum*-MSS and the two *S. europaea* samples (Table 2).

### 3.2. Bioactive Compound Concentrations and Antioxidant Capacity

Chlorophyll levels (CHL*a*, CHL*b*) in *S. europaea* and *A. macrostachyum*-MSS were significantly higher (more than 10 times) than in other collected glassworts (Table 3). Carotenoid concentrations were also elevated in *S. europaea*, particularly in the commercial farm sample (*S. europaea*-SC), followed by *S. fruticosa*-MSS. Total phenol (TP) content in *S. europaea* and *A. macrostachyum*-MSS was approximately 4.5 times higher than that in the other samples. Higher variability was detected for flavonoids (FL) with *A. macrostachyum*-MSS having the greatest amount. FL concentration was halved in *S. europaea* and decreased to about one-fourth in other *A. macrostachyum* genotypes, whereas negligible was the level of FL in *S. fruticosa*-MSS (Table 3). Anthocyanin content fluctuated, with clear differences observed between *S. fruticosa*-MSS (higher values) and *A. macrostachyum*-MSS (Table 3). Sterol concentration in *A. macrostachyum*-MSS was 3.6-fold higher than in *A. macrostachyum*-MA-BA, *S. fruticosa*-VL and *S. europaea*-SC. The glasswort of Laguna del Re Oasis (LRO) had very low sterol levels (Table 3).

The antioxidant capacity (Table 4), tested through three assays, was greater for *S. europaea*-SC followed by the other annual sample (*S. europaea*-LRO) and *A. macrostachyum*-MSS. Whereas lower values were measured in the other perennial plants belonging to the species *A. macrostachyum* and *S. fruticosa*, indistinctly from the native site.

## 4. Discussion

Glassworts, belonging to different genera of the Salicornioideae subfamily, can provide valuable functional foods and dietary supplements, raising the interest of the scientific community for their nutraceutical profile [6]. In the Mediterranean region, research has focused on the nutritional properties of these halophytes. In contrast, there is a relative lack of studies on the nutraceutical profile of both spontaneous and cultivated glassworts in Italy. This work explored their potential as a novel food source by examining both their functional properties and antinutritional traits, as well as how these factors change with species and provenance. To the best of our knowledge, this is the first study to report a comprehensive nutritional characterization of Italian glassworts, specifically from the Apulian region. A comparison was made with products from other Mediterranean areas.

### 4.1. Mineral Nutrients and Antinutrients

The mineral composition of halophytes is governed not only by environmental conditions but also by genotype [30]. In our study, Na and Cl were the dominant elements in the mineral profiles of all samples. Sodium concentrations ranged from 40 to 120 mg g^−1^ DW, while chloride concentrations ranged from 10 to 150 mg g^−1^ DW. Notable genotypic differences in sodium accumulation were observed. Among the perennial species, *Arthrocaulon macrostachyum* generally accumulated less sodium than *Sarcocornia fruticosa*. A clear exception was the *A. macrostachyum* sample from Monti d’Arena-Bosco Caggione (MA-BC), a site located immediately adjacent to the sea. In contrast, the annual species *Salicornia europaea* exhibited a pronounced tendency to accumulate sodium. This was highlighted by a significant disparity between samples from Laguna del Re Oasis (LRO), where *S. europaea* (*S. europaea*-LRO) had the highest sodium concentration and *A. macrostachyum* (*A. macrostachyum*-LRO) the lowest, despite similar soil characteristics and water salinity. *A. macrostachyum* showed low Na level even when grown close to a salt pan (*A. macrostachyum* from Margherita di Savoia Saltworks-MSS).

Halophytes employ various physiological mechanisms to manage salinity stress, including sodium exclusion at the root level, excretion via specialized structures (e.g., bladder cells), and inclusion-diluting absorbed sodium within succulent tissues [48]. Our data confirm that *S. europaea* is a sodium-including species, likely using Na for osmoregulation [49,50]. This obligate halophyte’s growth is stimulated by sodium availability [51], a trait further evidenced by the sample from the conventional Spirito Contadino (SC) farm, which was richer in Na than other samples from brackish areas. Furthermore, it demonstrates the potential role of this species in the remediation of Na- affected soils. Some studies have reported the introduction of halophytes, including *S. europaea* [33], in intercropping systems for salt soil remediation, demonstrating beneficial effects for companion crops due to a reduction of soil salinity [52].

The generally low sodium accumulation in *A. macrostachyum* was surprising, as this species is often suggested for phytoremediation due to its high biomass and NaCl uptake [14,53]. This discrepancy may be explained by the partitioning of sodium within the plant [54]. It should be considered that in the present work, only fresh stems representing the edible portion were analysed. Ramírez et al. [14] found that in the *Arthrocaulon* genus, sodium accumulates primarily in biologically less active tissues like the xylem, vascular bundles of lignified stems, and epidermis. Therefore, the sodium in our sampled *A. macrostachyum* was likely stored in roots and lignified parts, which were not analyzed.

Reported sodium concentrations in glassworts are highly variable, attributable to differing environments. Our values are consistent with studies on the same species from Spain and Portugal [1,11,55]. However, our measured sodium content in *A. macrostachyum* was lower than the 80–310 mg g^−1^ DW range reported by Ramirez et al. [14] across Mediterranean countries (sampling in both coastal and inland areas).

Sodium is an element of major concern for human health, due to its implications in vascular dysfunction. Indeed, the World Health Organization (WHO) and the European Food Safety Authority (EFSA) recommend not exceeding the sodium acceptable daily intake dose of 2.0 g [56,57]. Furthermore, to mitigate vascular dysfunction and the risk of high blood pressure, it is recommended to consume foods with a sodium-to-potassium (Na/K) ratio of less than 1 [20], as potassium is essential for the human body, reducing blood pressure and stroke incidence [34]. The samples of our survey showed a Na/K ratio ranging from 1.7 (on average) for *S. europaea*-SC (cultivated in non-saline soil) and *A. macrostachyum* collected at the MSS site to 5 for *A. macrostachyum*-MA-BC (grown close to the sea) (Table 2). However, despite their high sodium content, several studies have demonstrated that dried glassworts, when consumed as a salt substitute or ‘green salt’, can exhibit antihypertensive effects. This is attributed to their rich composition of bioactive compounds, including trans-ferulic acid, which has been found to reduce hypertension [12,16,58].

In the perennial species, chlorine levels were generally higher in *A. macrostachyum* than in *S. fruticosa*, suggesting differences in chlorine regulation mechanisms [59]. Moreover, the lower chlorine levels observed in *S. fruticosa*-MSS and *A. macrostachyum*-LRO, compared to other samples of the same species, may be due to specific conditions at the sampling sites. The occasional supply of fresh water could decrease the chlorine concentration in the soil, since both collecting sites (MSSa and LROa) were located where salt and fresh water converge, creating a unique ecological environment. As expected, low Cl content characterized the conventionally cultivated *S. europaea* (*S. europaea*-SC) (Table 2). The chlorine concentrations of our samples turn out to be slightly lower compared to literature findings [60,61].

Concentrations of nitrate, potassium, calcium and magnesium ranged from 0.4–2.1, 21–48, 13–35, and 6–15 mg g^−1^ DW, respectively. As expected, cultivated *S. europaea* accumulated more nutrients due to the application of fertilizers. Specifically, a greater amount of NO_3_ was detected in the sample harvested from the SC farm, where an intensive agricultural system is practiced. The elevated NO_3_ in wild *A. macrostachyum*-MA-BC was likely due to runoff from upstream cultivated land.

Within perennial species, the mineral content results in line with what is found in the literature for the same genera [11,31,62]. Instead, the mineral content of cultivated *S. europaea* is significantly higher than reported by other authors in cultivated plants of the same species [28,63], likely for differences in soil and crop management.

The nitrate accumulation in green leafy vegetables poses a potential food safety hazard, as the nitrates can be reduced into nitrites and nitrosamines (carcinogenic compounds [64]). Therefore, the European Community (EC 2023/915) has set thresholds for the commercialization of the most consumed leafy vegetables (lettuce, rocket, and spinach). Our samples had markedly lower values of these thresholds; at least 400 mg NO_3_ kg^−1^ FW both in *S. europaea*-SC and *A macrostachyum*-MA-BC. This suggests that these species are not prone to accumulating nitrate.

Oxalate, an antinutrient that can chelate minerals (e.g., calcium) and can lead to kidney stones [19]. To date, the mechanism regulating the synthesis of oxalates is still complex, as they act as plant mineral regulators, and the Chenopodioideae uses the complexation of free oxalic acid to maintain ionic equilibrium [29]. Some authors reported that the content of oxalates in halophytes increases as substrate salinity rises [65]. On the contrary, in *S. ramosissima* under saline irrigation conditions, it was observed that a rise in temperature and CO_2_ lowered oxalate levels [29]. Among our samples, *S. fruticosa* from Varano lake (*S. fruticosa*-VL) and *A. macrostachyum*-MSS had the highest value, likely due to site-specific environmental conditions. Accordingly, on wild plants of *Salicornia patula*, oxalates increased from 1.5 to 16.9 g kg^−1^ FW when moving from coastal to inland areas [21]. Except for these two, our samples showed lower oxalates than those observed in *S. ramosissima* (3.5 and 4 g kg^−1^ FW) in the unstressed plants [29]. Comparing our glassworts with vegetables well known for high oxalate contents, it emerged that only *S. fruticosa*-VL is richer in oxalate than spinach (6.0 g kg^−1^ FW), and in *A. macrostachyum*-MSS it is similar to Swiss chard (3.0 g kg^−1^ FW).

Iodine is an important micronutrient involved in the synthesis of thyroid hormones, which regulate various physiological processes [18]. There is some evidence of the potential for halophytes to be an interesting source of this oligoelement [17,47]. Iodine content varied significantly with sampling location, with the highest levels found in LRO, VL, and MA-BC sites. Changes in iodine concentration could be ascribed to site-specific variability in soil and/or sea spray contribution in accumulating this ion. The low iodine content of the product from the Spirito Contadino farm is attributable to a non-saline cultivation system.

Studies analyzing the iodine content in glassworts are limited, but they reported lower concentrations than those found in our samples. Oliveira-Alves et al. [55] in *S. ramosissima* found iodine values below 0.13 µg g^−1^ FW, and Duarte et al. [66] found values around 15 µg g^−1^ DW on *S. fruticosa*.

The recommended daily intake (RDI) of iodine for an adult is 150 µg, varying between 90 and 250 µg according to age, sex, and other aspects of life (such as pregnancy) [67]. The iodine RDI could be satisfied by consuming portions of glasswort close to 40–150 g, depending on the product’s provenance. The consumption of the iodine-rich glasswort (*S. europaea*-LRO, 349 µg per 100 g FW) does not pose a risk of toxic effects, as an intake of up to 1500 µg per day is accepted as safe [68].

This work did not include the analysis of heavy metals (HMs), which are a concern for human health [69]. Despite glassworts growing in degraded land potentially contaminated by HMs, they are reported to accumulate these elements mainly at the root level [14,24,63,70] without compromising their use as food. Nevertheless, the content of these elements deserves further research to shed light on both nutritional and antinutritional aspects of products.

### 4.2. Visual Quality, Nutritional and Antioxidative Traits

Consumer acceptance of glassworts is influenced by both visual appeals and nutritional quality. From a nutritional standpoint, glasswort is particularly appealing since it is characterized by a rich profile of antioxidant compounds, including pigments, phenolics, and sterols, which contribute to its overall health benefits.

Chlorophylls are primary determinants of the green colour in leafy vegetables, a key visual quality indicator for consumers [11]. Beyond aesthetics, they offer significant health benefits, such as anti-mutagenic, anti-cancer, and anti-inflammatory properties [4]. Early-life dietary intake may also help prevent obesity and reduce the risk of type 1 diabetes [71].

In our study, the annual species *S. europaea* and the perennial *A. macrostachyum* from the Margherita di Savoia Saltworks (MSS) site exhibited total chlorophyll concentrations an order of magnitude higher than other samples (Table 3), making them particularly visually appealing. Conversely, *S. fruticosa*, *A. macrostachyum*-LRO, and *A. macrostachyum*-MA-BC had lower chlorophyll levels, which could suggest a protective response to environmental stresses. These conditions are reported to lead to excessive plant energy exposure, exceeding its capacity to use light effectively in photosynthesis [72]. In these samples, chlorophyll degradation and a reduced Chl*a*/Chl*b* ratio (averaging 1.3 vs. 2.8) likely served as a protective mechanism [73], a phenomenon also reported in other halophytes, including *Sarcocornia* [31]. This hypothesis is supported by their consistently higher anthocyanin concentrations (Table 3), known photosystem protectants. Our chlorophyll values are generally consistent with some literature [22], though lower than values reported for *S. ramosissima* from Portugal [11].

Carotenoids are pigments with potent antioxidant and immunological properties [11]. Lutein and β-carotene, for example, support vision, protect against macular degeneration, maintain skin health, and combat free radicals [74]. The European Food Safety Authority [75] guidelines consider carotenoids (like β-carotene) only as a supplement to meet vitamin A requirements.

In plants, they absorb excess light energy, prevent ROS formation, and protect the photosynthetic apparatus from salt stress in a species-dependent manner [72,76]. A rise in carotenoids was observed in *A. macrostachyum* and *S. fruticosa* with the increase of NaCl in the nutrient solution, while *S. europaea* showed the opposite trend [76]. Notably, the *S. europaea* sample from the non-saline Spirito Contadino (SC) farm was richest in carotenoids (Table 3), aligning with findings that this species may reduce carotenoid accumulation under salinity [76]. Our carotenoid values are similar to those reported for Mediterranean glassworts from different areas of Portugal, Spain, and Tunisia [22,31,54] but lower than those found in plants from Portugal’s Algarve region [11].

Phenolic compounds, particularly flavonoids and anthocyanins, are strong radical scavengers [64] that confer anti-inflammatory, antiviral, and antibacterial benefits [77,78]. Adequate anthocyanin intake is linked to improved neurological and cardiovascular health [79].

In plants, phenol synthesis is a primary defense mechanism against oxidative stress [80]. Anthocyanins, specifically, filter light to prevent ROS production [47].

In halophytes, phenolic compounds are known to help alleviate the harmful effects of harsh environmental conditions. As soil salinity or air temperature rose, their biosynthesis increase was reported, though the response was genotype specific [66]. In glassworts, when salt stress was applied under a greenhouse-soilless system, moderate salt concentration at the root level led to an improvement of TP in *Salicornia* and *Sarcocornia* species [28]. Moreover, *A. macrostachyum* showed significant increases in phenols under moderate/high salinity, while *S. europaea* levels often remained constant [76]. When comparing wild and soilless cultivated *S. fruticosa*, it was suggested that phenol content was positively affected by stresses encountered in wild conditions, beyond just salinity [31], such as drought and unbalanced nutrient availability. Therefore, strong interaction between genotypes and various abiotic/biotic stress is expected regarding phenol content.

We observed clear differences in total phenol (TP) content. Specifically, *A. macrostachyum* collected near saltworks (MSS) and *S. europaea* cultivated under both saline (LRO) and non-saline (SC) conditions showed significantly more TP compared to other samples. Similarly, a slight improvement in flavonoid content was found in these same plants (Table 3). The likely explanation for this overproduction of TP, as an antioxidative response, is the reduced ability of these plants to prevent oxidative damage by decreasing chlorophylls, a phenomenon observed in other samples. It can be argued that *S. europea* appears predisposed to synthesizing phenols as a mechanism to mitigate environmental stress without involving photosynthetic pigments. It’s noteworthy that the increase in TP was also observed in plants grown on non-saline commercial farms, suggesting growth-stressing conditions even there. In *S. europaea* samples, a further possible explanation for the higher TP levels could be related to the harvest period. Both cultivated glassworts were collected during the hottest months (August-September), while the wild samples were collected during the spring period. This seasonal variation may have influenced the plants’ secondary metabolite production, including phenolics. An improvement of Folin-Ciocalteau TPs was observed in *S. ramossissima* harvested in July compared to May in Portugal [22]. The *A. macrostachyum* samples near saltworks (*A. macrostachyum*-MSS) represent a strong example of genotype-environment interaction. This is evident when considering its lower TP and chlorophyll contents compared to the same species collected from other areas (*A. macrostachyum*-MA-BC; *A. macrostachyum*-LRO) (Table 3), suggesting a varied response to the environmental pressure across the sites.

As expected, TP content widely varies among Mediterranean glassworts. Our TP values for *S. europaea* and *S. fruticosa* align with those from cultivated plants [1,28] but are generally lower than those from wild plants in Portugal, Turkey, and Spain [11,22,31,81]. *A. macrostachyum* values were consistent with some studies [24,82] but lower than others [11,81,83]. Flavonoid content was slightly higher across our samples compared to literature values for the *Sarcocornia*, *Salicornia*, and *Arthrocaulon* genera [11,31,81,82]. Anthocyanin levels in *S. europaea* aligned with previous reports, while those in *S. fruticosa* were slightly higher [84]; no studies on anthocyanin content in *A. macrostachyum* were found.

Sterols, naturally present in plants, have anti-tumorigenic effects and prevent free radical damage by lowering blood cholesterol, resulting in reduced coronary heart disease [85]. A lowering effect of low-density lipoprotein cholesterol level in blood is reported for phytosterols [86]. In plants, they are essential in embryogenesis processes and for cellular membrane fluidity and permeability [87].

The biosynthesis of these compounds is influenced by both soil and climatic factors. For instance, Pavlík et al. [88] observed increases in sterol content in maize plants with improved nitrogen availability. Roche et al. [87,89] also provided evidence that environmental conditions like higher temperature and lower rainfall positively affected phytosterol content in sunflowers and safflowers. This underscores the role of phytosterols in maintaining proton balance within cells, regulating their activity, particularly in terms of membrane fluidity and permeability, helping the cells adapt to oxidative processes occurring in unfavorable environmental conditions [87].

Research on sterol changes in halophytes is limited. Samples of *Salicornia perennis* collected from the coastal lagoon of Rio Aveiro (Portugal) exhibited sterol concentrations of 210 mg g^−1^ DW in sites characterized by lower salinity and more frequent flooding. However, in areas with elevated salinity and drier conditions, these concentrations soared to 600 mg g^−1^ DW. It was suggested that the observed increase in sterols helped protect cell membrane lipids, thereby enhancing membrane resistance [90]. Also in *Spartina patens*, Rozentsvet et al. [91] reported that an increment in plasmalemma sterols was connected to rising NaCl concentrations in the growth medium. These studies highlight the role of sterols in maintaining membrane integrity during salt stress, even at the cost of increased membrane permeability.

As reported for the other compounds mentioned so far, sterol content is significantly influenced by plant location, with the *A. macrostachyum*-MSS sample significantly exceeding other glassworts. This finding may be attributed to the characteristics of its sampling area, which is marked by a lack of water during certain times of the year. The presence of this species on a poorly watered site validates its capability to thrive in arid conditions. It should be noted that in the case of *S. fruticosa* collected from the same area (MSS), the exact spot was near the stream mouth, in a flooded area. Most of our samples had sterol values similar to *S. ramosissima* [92], while *A. macrostachyum*-MSS reached levels comparable to *Salicornia perennis* under moderate stress [90]. Compared to commonly consumed vegetables, our glassworts are similar to Chinese cabbage and carrot [85], with *A. macrostachyum*-MSS overcoming the sterol content of broccoli, Brussels sprouts, and spinach, and matching that of peas and black olives [85,93]. The high levels present in this population suggest *A. macrostachyum* could be a promising candidate for further investigation as a source of natural phytosterols. However, it is crucial to note that this study quantifies content rather than bio-efficacy; thus, while the potential for cholesterol-lowering claims exists, specific health benefits would require validation through clinical studies.

The antioxidant activity (AC) of complex systems (such as plant extracts) cannot be evaluated only with one specific assay, because it involves distinct radicals or antioxidant sources [94]. Therefore, the antioxidant activity was assessed by three complementary and widely used assays, including antioxidant reaction with a Fe (III) complex (FRAP), reaction with organic radical cation (ABTS), and reaction with organic radical (DPPH). Antioxidant activity increases in conjunction with stress conditions and is typically related to the production of secondary metabolism compounds.

The results of the three assays showed a similar trend, as also noted by Rumpf et al. [94] on different vegetable aqueous extracts. The hydrophilic antioxidant capacity (HA: from phenolics, flavonoids, anthocyanins, vitamin C) was the dominant contributor, vastly exceeding the lipophilic capacity (LA: from carotenoids, chlorophylls, sterols).

The two annual glasswort samples had improved antioxidant activity compared with the perennial species. An exception was found in the *A. macrostachyum*-MSS sample, which exhibited antioxidant levels that were comparable to those of *S. europaea*. It is plausible to hypothesize that the observed increase in antioxidant activity was driven by higher phenolics; however, it is important to note that we did not investigate other hydrophilic compounds, such as ascorbic acid, which might also have played a significant role in influencing hydrophilic antioxidant capacity. Regarding the *Arthrocaulon* of MSS, the higher AC confirmed a stronger site-specific stress pressure, leading the plant to exhibit an elevated antioxidant response, even when compared to other samples of the same species. In general, the correlation between the hydrophilic antioxidant capacity ranged from 0.61 to 0.76 for total phenols and from 0.59 to 0.84 for flavonoids (Appendix A). The highest values detected for the DPPH test underlined a strong positive with phenols.

In *Salicornia ramosissima* and *Sarcocornia perennis*, Hupel et al. [95] and Antunes et al. [22] reported that in response to oxidative stress, the amount of carotenoids and phenols increased, which resulted in a similar magnitude of changes in antioxidant capacity.

### 4.3. Principal Component Analysis

Principal component analysis (PCA) was employed to visualize the key biochemical traits differentiating glasswort species and their growing environments across Apulia region (Figure 2A, B). The first three principal components (PCs), with eigenvalues >1, collectively explained 88% of the total variance in the dataset. The biplot of the first two components (PC1 and PC2), which accounted for 75.5% of the variance, revealed distinct groupings (Figure 2A). PC1 was positively correlated with essential minerals (K, Mg, Ca, Cl), photosynthetic pigments (chlorophylls, carotenoids), and phenolic compounds. PC2 was positively associated with anthocyanins, sodium, and nitrate, and negatively correlated with chlorine, sterols, and flavonoids. A third component, PC3, was primarily defined by a strong positive correlation with iodine and a weaker one with Na, and a negative correlation with sterols and nitrate (Figure 2B).

The first two PCA components (Figure 2A) clearly differentiated the two annual *S. europaea* samples and the perennial *A. macrostachyum*-MSS from the other perennial species. The *S. europaea* samples, clustering in the upper right quadrant, were characterized by their high mineral and chlorophyll content. This aligns with ANOVA results and suggests a strong influence of nutrient availability, likely from fertilization practices at both the non-saline (SC) and saline (LRO) cultivation sites. Notably, *S. europaea*-LRO’s accumulation of essential cations (K, Ca, Mg) despite high soil NaCl competition highlights its efficient ion selectivity.

In contrast, *A. macrostachyum*-MSS, located in the lower right quadrant, was distinguished by its high chlorine content and bioactive profile, rich in sterols and flavonoids but poorer in anthocyanins. This clustering suggests that an excessive soil concentration and plant uptake of chlorine could have provoked a response to the oxidative stress in this sample involving sterols and flavonoids biosynthesis [96].

The remaining perennial *Sarcocornia* and *Arthrocaulon* samples clustered on the left side of PC1, showing a strong positive association with anthocyanins. This contrast with the *S. europaea* and *A. macrostachyum*-MSS clusters confirms that different halophyte species employ distinct biochemical strategies to cope with environmental stress and that the specific environmental condition effect may also mediate genotype response.

Furthermore, the importance of sodium for glasswort physiology was underscored by its significant loading on both PC2 and PC3. The latter was particularly useful in separating samples based on their iodine content, with *S. europaea*-LRO, *A. macrostachyum*-LRO, and *S. fruticosa*-VL forming a group characterized by higher accumulation. Conversely, the same perennial species collected at the MSS site, along with *S. europaea*-SC, were the poorest in iodine. This pattern can be attributed to the site-specific bioavailability of this ion. In the *S. europaea*-SC sample, it was expected because of its cultivation on a conventional farm. For the wild glassworts, it can be explained by *A. macrostachyum* was found in a location shielded from frequent marine flooding, and *S. fruticose* was located at the mouth of brackish water. Points may be potentially less rich in iodine, which could have led to less accumulation of this element in plants.

## 5. Conclusions

This study provides a comprehensive biochemical and nutritional profile of both wild and cultivated glassworts from the Apulia region, offering critical insights for their development as a novel food source. Our results confirm that genotypes (*Salicornia europaea*, *Arthrocaulon macrostachyum*, *Sarcocornia fruticosa*) are the primary driver of mineral and bioactive compound composition, but their expression may be deeply modulated by environmental conditions, leading to significant site-specific variation.

In support of the initial aim, we found that these halophytes possess great nutritional potential. The annual species *S. europaea* consistently demonstrated promising nutraceutical traits, accumulating high levels of essential minerals (K, Ca, Mg), carotenoids, and phenolic compounds, even under conventional non-saline conditions. This confirms its suitability as a nutrient-dense, sustainable crop. Furthermore, all species served as an excellent source of iodine, with a single portion capable of meeting the recommended daily intake without risk of toxicity. However, the elevated levels of Na and oxalates in some samples may raise concerns regarding their dietary suitability, highlighting the importance of assessing product provenance. Implementing appropriate agricultural practices could help manage anti-nutritional compounds, thereby reducing the sodium-to-potassium (Na/K) ratio and limiting oxalate accumulation.

Although this study provides a snapshot of metabolite profiles during one spring–summer period, we recognize that these profiles may be subject to seasonal and interannual variation. Future studies involving multi-seasonal sampling would be valuable to elucidate the full dynamic range of metabolic responses in these halophyte species.

This research reinforces the existing literature on the value of these halophytes but moves the discussion forward by highlighting the critical importance of provenance and genotype selection. Future efforts must focus on elucidating the specific mechanisms behind the observed genotype-environment interactions, particularly the physiological strategies governing ion uptake and regulation, and antioxidant production. This will be essential for standardizing product quality, ensuring food safety, and ultimately leveraging the full potential of glassworts as a sustainable and nutritionally enriched food source for a resilient agricultural system.

## Figures and Tables

**Figure 1 foods-14-03433-f001:**
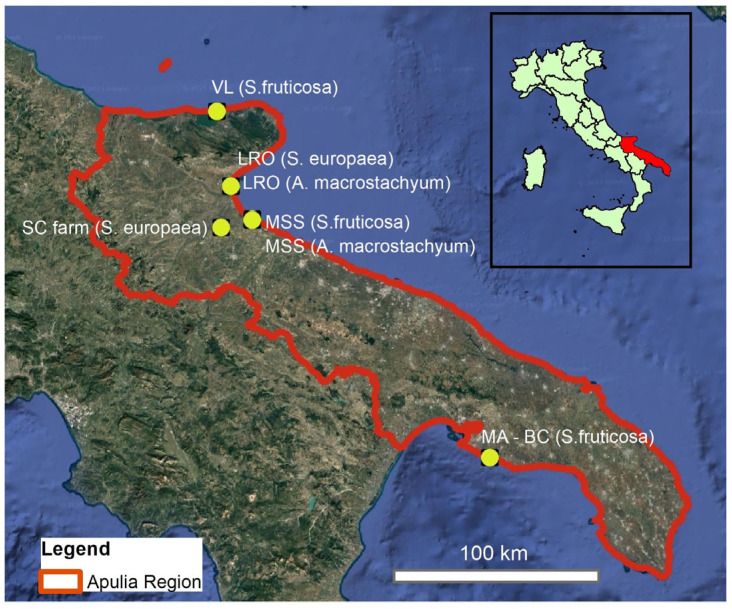
Geographic location of the sampling sites. The base satellite image, retrieved from Google Earth Satellite, was georeferenced and annotated to indicate sample collection points. Corresponding site identifiers are detailed in Table 1.

**Figure 2 foods-14-03433-f002:**
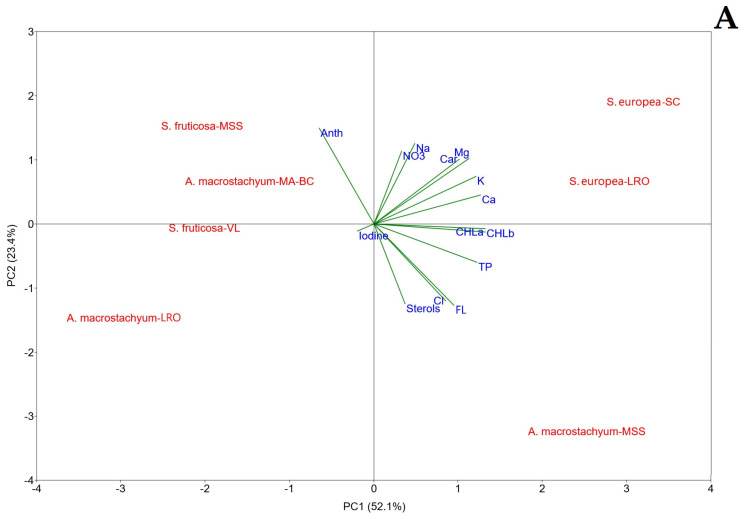
Principal Component Analysis bi-plot ((**A**)—PC1 vs. PC2; (**B**)—PC1 vs. PC3) reporting the minerals and biochemicals. Specifically: Nitrate (NO_3_), Sodium (Na), Potassium (K), Chlorine (Cl), Calcium (Ca), Magnesium (Mg), Iodine, chlorophyll *a*, and *b* (CHLa, CHLb), carotenoids (Car), Anthocyanins (Anth), Total Phenols (TP), Flavonoids (FL), and Sterols. Please see Table 1 for sample identification.

**Table 1 foods-14-03433-t001:** Sampling areas, taxonomic classification and sample identification of the collected glassworts.

Sampling Area	Taxonomic Classification	Sample Identification
Margherita di Savoia Saltworks (MSS), site a	*Sarcocornia fruticosa* L.	*S. fruticosa*-MSS
Margherita di Savoia Saltworks (MSS), site b	*Arthrocaulon macrostachyum* Moric.	*A. macrostachyum*-MSS
Laguna del Re Oasis (LRO), site a	*Arthrocaulon macrostachyum* Moric.	*A. macrostachyum*-LRO
Laguna del Re Oasis (LRO), site b	*Salicornia europaea* Auct.	*S. europaea*-LRO
Varano lake (VL)	*Sarcocornia fruticosa* L.	*S. fruticosa*-VL
Monti d’Arena-Bosco Caggione (MA-BC)	*Arthrocaulon macrostachyum* Moric.	*A. macrostachyum*-MA-BC
Spirito Contadino farm (SC)	*Salicornia europaea* Auct.	*S. europaea*-SC

**Table 2 foods-14-03433-t002:** Dry mass and mineral contents of annual and perennial glassworts collected from different areas of the Apulia region.

Glasswort Sample	Dry Mass	Na	K	Mg	Ca	Cl	NO_3_	C_2_O_4_	Na/K	I
	g·kg^−1^ FW									µg 100 g^−1^ FW
*S. europaea*-LRO	178.3 b ^(1)^ (±3.9)	20.9 a (±0.8)	8.7 ab (±0.6)	2.2 ab (±0.1)	6.1 ab (±0.4)	21.9 b (±1.0)	0.08 b (±0.00)	0.15 d (±0.01)	2.9 bc (±0.2)	349.4 a (±21.0)
*S. europaea*-SC	194.2 b (±9.8)	14.8 abc (±1.0)	9.0 a (±0.6)	2.8 a (±0.3)	6.9 a (±0.8)	15.4 c (±1.0)	0.39 a (±0.06)	0.78 cd (±0.09)	1.6 c (±0.1)	96.5 c (±8.8)
*S. fruticosa*-MSS	187.0 b (±6.8)	13.4bcd (±1.0)	4.0 bc (±0.4)	1.4 bc (±0.1)	4.2 abc (±1.0)	1.9 e (±1.8)	0.07 b (±0.04)	0.01 d (±0.00)	3.8 ab (±0.7)	104.8 c (±9.3)
*S. fruticosa*-VL	189.0 b (±5.5)	14.9 abc (±0.2)	4.1 abc (±0.3)	1.3 c (±0.1)	2.5 bc (±1.0)	12.9 cd (±0.7)	0.09 b (±0.01)	6.57 a (±0.47)	3.7 ab (±0.2)	212.0 abc (±13.2)
*A. macrostachyum*-MA-BC	187.0 b (±5.7)	17.1 ab (±0.4)	3.4 c (±0.3)	1.2 c (±0.04)	3.6 abc (±0.4)	21.7 b (±0.7)	0.40 a (±0.01)	1.47 c (±0.06)	5.2 a (±0.4)	223.7 abc (±16.1)
*A. macrostachyum*-LRO	104.6 c (±5.3)	6.9 d (±0.3)	1.9 c (±0.2)	0.8 c (±0.04)	1.8 c (±0.3)	8.9 d (±0.5)	n.d. ^(3)^	0.79 cd (±0.12)	4.0 ab (±0.3)	277.4 ab (±9.8)
*A. macrostachyum*-MSS	242.2 a (±5.1)	9.8 cd (±0.4)	5.4 abc (±0.7)	1.5 bc (±0.01)	5.7 ab (±0.5)	37.5 a (±0.2)	0.11 b (±0.00)	2.96 b (±0.27)	1.9 c (±0.3)	112.1 bc (±2.9)
Significance ^(2)^	***	***	***	***	***	***	***	***	***	***

^(1)^ Means (±standard error) in columns not sharing the same letters are significantly different according to Tukey’s HSD test at α ≤ 0.05; ^(2)^ ***: significant at *p* < 0.001. Values are the means of eighteen subsamples (n = 18); ^(3)^ n.d.: <LOD = 0.45 mg L^−1^.

**Table 3 foods-14-03433-t003:** Pigment and antioxidant compound contents of annual and perennial glassworts collected from different areas of the Apulia region. Specifically: chlorophyll *a*, *b*, and total (CHL*a*, *b*, t), carotenoids (Car), Anthocyanins (Anth), Total Phenols (TP), Flavonoids (FL), and Sterols.

Glasswort Sample	CHL*a*	CHL*b*	CHLt	Car	TP	FL	Anth	Sterols
	µg g^−1^ FW			mg ßc.e. ^(2)^ 100 g^−1^ FW	mg g.a.e. ^(2)^ 100 g^−1^ FW	mg q.e. ^(2)^ 100 g^−1^ FW	mg c.g.e ^(2)^ 100 g^−1^ FW	mg s.e. ^(2)^ 100 g^−1^ FW
*S. europaea*-LRO	134.4 a ^(1)^ (±4.7)	49.5 a (±2.0)	180.8 a (±6.4)	4.1 b (±0.20)	258.3 a (±7.7)	31.7 bc (±0.00)	0.7 bc (±0.10)	7.4 c (±0.5)
*S. europaea*-SC	127.0 a (±21.3)	43.0 a (±8.8)	167.7 a (±29.3)	6.2 a (±0.30)	192.9 a (±22.3)	39.3 b (±2.20)	1.0 abc (±0.20)	12.2 bc (±1.4)
*S. fruticosa*-MSS	11.5 b (±0.6)	8.0 b (±1.2)	18.7 b (±1.6)	3.5 b (±0.03)	19.3 b (±2.0)	0.3 e (±0.02)	1.6 a (±0.03)	n.a. ^(3)^
*S. fruticosa*-VL	7.8 b (±0.2)	6.3 b (±0.1)	13.4 b (±0.3)	1.7 c (±0.02)	26.6 b (±1.0)	26.2 cd (±0.60)	1.1 abc (±0.02)	17.0 b (±0.9)
*A. macrostachyum*-MA-BC	12.3 b (±0.2)	10.8 b (±0.2)	21.8 b (±0.3)	1.3 c (±0.05)	78.4 b (±3.5)	22.0 cd (±1.70)	1.4 ab (±0.03)	15.3 bc (±1.0)
*A. macrostachyum*-LRO	6.9 b (±1.2)	4.7 b (±1.0)	11.0 b (±2.1)	1.2 c (±0.10)	83.7 b (±6.6)	18.0 d (±0.90)	0.6 bc (±0.20)	7.3 c (±0.9)
*A. macrostachyum*-MSS	119.7 a (±3.3)	41.5 a (±0.7)	159.0 a (±3.8)	2.8 bc (±0.10)	256.2 a (±7.3)	73.9 a (±2.00)	0.4 c (±0.10)	53.3 a (±10.0)
Significance ^(4)^	***	***	***	***	***	***	***	***

^(1)^ Means (±standard error) in columns not sharing the same letters are significantly different according to Tukey’s HSD test at α ≤ 0.05; ^(2)^ ßc.e. = ß-caroten equivalent; g.a.e = gallic acid equivalent; q.e = quercetin equivalent; c.g.e = cyanidin-3-glucoside equivalent; s.e. = stigmasterol equivalent; ^(3)^ n.a. = not analyzed; ^(4)^ *** = significant at *p* < 0.001. Values are the mean of eighteen subsamples (n = 18).

**Table 4 foods-14-03433-t004:** Antioxidant capacity of annual and perennial glassworts collected from different areas of the Apulia region tested through DPPH, ABTS and FRAP assays.

Glasswort Sample	DPPH	ABTS			FRAP		
		HA ^(2)^	LA ^(2)^	Total	HA ^(2)^	LA ^(2)^	Total
	µmol T.E. g^−1^ FW ^(3)^						
*S. europaea*-LRO	8.2 b ^(1)^ (±0.2)	12.4 c (±0.4)	0.60 a (±0.02)	13.0 c (±0.4)	19.5 b (±0.7)	n.d.	19.5 b (±0.7)
*S. europaea*-SC	13.0 a (±0.5)	22.9 a (±0.9)	0.73 a (±0.05)	23.7 a (±1.0)	71.6 a (±1.2)	n.d.	71.6 a (±1.2)
*S. fruticosa*-MSS	3.6 d (±0.1)	6.9 d (±0.2)	n.d. ^(4)^	6.9 d (±0.2)	5.8 d (±0.2)	0.26 b (±0.03)	6.2 d (±0.2)
*S. fruticosa*-VL	5.8 c (±0.2)	7.3 d (±0.3)	n.d.	7.3 d (±0.3)	8.4 cd (±0.4)	0.43 a (±0.03)	8.8 cd (±0.4)
*A. macrostachyum*-MA-BC	2.6 d (±0.1)	5.4 d (±0.4)	n.d.	5.4 d (±0.4)	5.0 d (±0.2)	0.33 ab (±0.02)	5.3 d (±0.1)
*A. macrostachyum*-LRO	4.1 d (±0.2)	4.4 d (±0.2)	n.d.	4.4 d (±0.2)	7.8 d (±0.4)	0.40 ab (±0.04)	7.5 d (±0.4)
*A. macrostachyum*-MSS	9.4 b (±0.2)	17.8 b (±1.2)	0.27 b (±0.02)	18.1 b (±1.2)	14.3 bc (±0.6)	0.10 c (±0.00)	14.4 bc (±0.6)
Significance ^(5)^	***	***	***	***	***	***	***

^(1)^ Means (±standard error) in columns not sharing the same letters are significantly different according to Tukey’s HSD test at α ≤ 0.05; ^(2)^ HA = hydrophilic fraction of antioxidants; LA = lipophilic fraction of antioxidants; ^(3)^ T.E. = Trolox equivalent; ^(4)^ n.d.: <LOD = 0.15 and 11.7 µmol T.E. for LA-ABTS and LA-FRAP, respectively; ^(5)^ *** = significant at *p* < 0.001. Values are the mean of eighteen subsamples (n = 18).

## Data Availability

The data supporting the findings of this study are available from the corresponding authors upon request.

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
