# Peer review of "Nutritional Characterization of Annual and Perennial Glassworts from the Apulia Region (Italy)"

_foods, 2025, doi:10.3390/foods14193433_

Round 1
Reviewer 1 Report
Comments and Suggestions for Authors
General Evaluation
The manuscript by Duri et al, presents an original and comprehensive biochemical characterization of annual (Salicornia europaea) and perennial (Sarcocornia fruticosa, Arthrocaulon macrostachyum) glassworts collected and cultivated in the Apulia region of Italy. The study is timely, addressing the rising interest in halophytes as climate-resilient crops and their integration into sustainable diets and functional foods. The experimental design is solid, encompassing both wild and cultivated materials from multiple sites, while the analytical coverage is broad, spanning mineral composition, bioactive compounds, pigments, sterols, and antioxidant activity via complementary assays.
Strengths
-Novel Contribution: This is the first systematic report of the biochemical profile of Italian glassworts, filling both a geographical and scientific gap. Notably, this appears to be the first documentation of iodine and sterols in halophytes from the Apulia region.
-Breadth of Analysis: The integration of mineral and phytochemical traits, combined with antioxidant activity assessed through three independent methods, provides a multidimensional view of nutritional potential.
-Comparative Approach: By evaluating annual and perennial taxa across different habitats and cultivation systems, the study sheds light on genotype–environment interactions.
-Sampling Context: Figure 1 effectively situates the sampling sites, enhancing ecological interpretation.
-Relevance to Sustainability: Framing halophytes as candidate crops for saline and marginal lands underscores their potential contributions to sustainable agriculture and climate-resilient food systems.
Points for Revision
The authors include the species Salicornia europea auct. (synonyms: Salicornia ramosissima Auct., Salicornia patula Duval-Jouve, Salicornia perennans Willd. subsp. perennans) [8,9]. However, the cited bibliographical references do not include Salicornia europea within the Flora of Italy. It is true that the material used in the essays corresponds to material from the Aurora nursery ‘Aurora’ (San 147 Nicandro Garganico, FG - Italy). I think the authors should clarify taxonomically whether they consider the species S. europea as the only species in the Italian flora.
-Food Safety Considerations: The absence of data on potentially toxic elements (Cd, Pb, As, Hg) limits the translational value for human consumption. A discussion on toxic elements and the phytoremediation potential of halophytes should be included.
Reference 23, in line 92, requires proper contextualization, and it is essential to incorporate the works by Sánchez-Gavilán et al. The work authors cited is a reference about mineral content, not bioactive compounds, cited both correctly.
Sánchez-Gavilán, I., Rufo, L., Rodríguez, N., & de la Fuente, V. (2021). On the elemental composition of the Mediterranean euhalophyte Salicornia patula Duval-Jouve (Chenopodiaceae) from saline habitats in Spain (Huelva, Toledo and Zamora). Environmental Science and Pollution Research, 28(3), 2719-2727.
Sánchez-Gavilán, I., Ramírez, E., & de La Fuente, V. (2021). Bioactive compounds in Salicornia patula Duval-Jouve: A Mediterranean edible euhalophyte. Foods, 10(2), 410.
-Temporal Scope: Sampling was limited to a single season (spring–summer 2023). Since seasonal and interannual variability can strongly affect halophyte metabolite profiles, specially salt stress, this limitation should be acknowledged in the manuscript.
-Dietary Interpretation: While iodine is quantified and represents a significant novel finding, the dietary implications of other minerals and bioactives are not addressed in relation to dietary reference values, bioavailability, or realistic intake levels.
-Data Presentation: Although statistically robust, the extensive tables could be complemented with graphical visualizations to highlight nutritionally relevant contrasts (e.g., Na/K ratios, antioxidant potential compared to other edible halophytes).
-Conceptual Framing: The manuscript leans toward a biochemical characterization rather than a nutritional interpretation. Clarifying this scope would better align the study with its stated objectives and avoid overstating dietary relevance.
-Conclusion
This is a rigorous and valuable contribution to halophyte biochemistry and sustainability research. Its major strengths lie in its comprehensive profiling of underexplored taxa and its contextualization within the framework of climate-resilient agriculture. To maximize its impact, the manuscript should (i) integrate food safety considerations, (ii) strengthen the dietary and comparative interpretation by referencing additional Mediterranean data, and (iii) improve the visualization of results.
Recommendation: Minor Revision.
Author Response
|
Reviewer 1 |
|
1 |
The authors include the species Salicornia europea auct. (synonyms: Salicornia ramosissima Auct., Salicornia patula Duval-Jouve, Salicornia perennans Willd. subsp. perennans) [8,9]. However, the cited bibliographical references do not include Salicornia europea within the Flora of Italy. It is true that the material used in the essays corresponds to material from the Aurora nursery ‘Aurora’ (San Nicandro Garganico, FG - Italy). I think the authors should clarify taxonomically whether they consider the species S. europea as the only species in the Italian flora. |
We thank the reviewer for highlighting the recent taxonomic treatment by Bartolucci et al. (2024), where Salicornia europaea auct. is synonymized with Salicornia perennans Willd. subsp. Perennans (see Appendix S2). While we acknowledge the updated nomenclature, we have chosen to use the name “S. europaea” in this manuscript to maintain consistency with its common usage in the existing agronomic and food science literature cited throughout our work, thereby ensuring clarity for our target audience. |
2 |
-Food Safety Considerations: The absence of data on potentially toxic elements (Cd, Pb, As, Hg) limits the translational value for human consumption. A discussion on toxic elements and the phytoremediation potential of halophytes should be included. |
We thank the reviewer for raising the important point regarding heavy metal (HM) content. We agree that this is a critical aspect for any food source, especially one grown in potentially marginal environments. We are aware that food contaminated by heavy metals can lead to serious health problems, and that these halophytes can accumulate toxic heavy metals (such as Cd, Pb, As, Hg). However, HM accumulations in glasswort seem to occur mainly at the root level, preserving food purposes as reported in many studies (Sánchez-Gavilán 2021; García Caparros et al., 2022; Lopes et al., 2023, Ramírez et al., 2024). Anyway, we agree with the referee about the importance of mentioning heavy metals in more detail in the text, and we have therefore added a discussion at lines 643-648 “This work did not include the analysis of heavy metals (HMs), which are a concern for human health [69]. Despite glassworts growing in degraded land potentially contam-inated by HMs, they are reported to accumulate these elements mainly at the root level [14,24,63,70] without compromising their use as food. Nevertheless, the content of these elements deserves further research to shed light on both nutritional and antinutritional aspects of products.” |
3 |
Reference 23, in line 92, requires proper contextualization, and it is essential to incorporate the works by Sánchez-Gavilán et al. The work authors cited is a reference about mineral content, not bioactive compounds, cited both correctly.
Sánchez-Gavilán, I., Rufo, L., Rodríguez, N., & de la Fuente, V. (2021). On the elemental composition of the Mediterranean euhalophyte Salicornia patula Duval-Jouve (Chenopodiaceae) from saline habitats in Spain (Huelva, Toledo and Zamora). Environmental Science and Pollution Research, 28(3), 2719-2727.
Sánchez-Gavilán, I., Ramírez, E., & de La Fuente, V. (2021). Bioactive compounds in Salicornia patula Duval-Jouve: A Mediterranean edible euhalophyte. Foods, 10(2), 410. |
We thank the reviewer for their careful review. We apologize for the error; we mistakenly cited the wrong paper. We have corrected this by citing Sánchez-Gavilán, I., Ramírez, E., & de La Fuente, V. (2021). Bioactive compounds in Salicornia patula Duval-Jouve: A Mediterranean edible euhalophyte. Foods, 10(2), 410.
Please see reference 26 |
4 |
-Temporal Scope: Sampling was limited to a single season (spring–summer 2023). Since seasonal and interannual variability can strongly affect halophyte metabolite profiles, specially salt stress, this limitation should be acknowledged in the manuscript. |
We thank the reviewer for this insightful comment. We agree that seasonal and interannual variability are crucial factors influencing metabolite profiles in halophytes. We have now explicitly acknowledged this limitation in the Conclusion section of the manuscript (as shown in the new sentence below). We believe this addition provides important context for interpreting our results and suggests a valuable direction for future research. Added in the Conclusion section lines 881- 884: “Although this study provides a snapshot of metabolite profiles during one spring–summer period, we recognize that these profiles may be subject to seasonal and interannual variation. Future studies involving multi-seasonal sampling would be valuable to elucidate the full dynamic range of metabolic responses in these halophyte species.” |
5 |
-Dietary Interpretation: While iodine is quantified and represents a significant novel finding, the dietary implications of other minerals and bioactives are not addressed in relation to dietary reference values, bioavailability, or realistic intake levels. |
Thank you for the question. We take home this advice for future research specifically focused on these aspects, but the primary focus of this work is the biochemical characterization of glasswort samples from the Apulian region, which has never been done before. By the way, targeted dietary considerations have been discussed specifically for compounds of paramount relevance to human health - namely, sodium and iodine- given their significant abundance in the analyzed samples and their direct implications for nutritional intake. |
6 |
-Data Presentation: Although statistically robust, the extensive tables could be complemented with graphical visualizations to highlight nutritionally relevant contrasts (e.g., Na/K ratios, antioxidant potential compared to other edible halophytes). |
Thank you for your interesting question. We agree with the reviewer that a comparison with other halophytes would be interesting, but it would need to be contextualized and would imply an extensive analysis behind the main aim of the study |
7 |
-Conceptual Framing: The manuscript leans toward a biochemical characterization rather than a nutritional interpretation. Clarifying this scope would better align the study with its stated objectives and avoid overstating dietary relevance. |
We agree with the reviewer and have clarified the scope throughout the manuscript to emphasize that this is primarily a biochemical characterization study. Any nutritional discussion is now explicitly framed as a secondary consideration, focused only on the implications of the most salient findings (e.g., Na/K ratio, iodine content) for future research, not dietary recommendations. |
8 |
-Conclusion This is a rigorous and valuable contribution to halophyte biochemistry and sustainability research. Its major strengths lie in its comprehensive profiling of underexplored taxa and its contextualization within the framework of climate-resilient agriculture. To maximize its impact, the manuscript should (i) integrate food safety considerations, (ii) strengthen the dietary and comparative interpretation by referencing additional Mediterranean data, and (iii) improve the visualization of results. |
Thank you for the positive comments.
We agree with the referee on the importance of a comparative evaluation of these new products in terms of their safety and dietary value. We believe that further review work could address this scope. |

Reviewer 2 Report
Comments and Suggestions for Authors
For detailed modification suggestions, please refer to the attachment.

Author Response
|
Reviewer 2 |
|
1 |
This manuscript provides a comprehensive nutritional and biochemical characterization of annual (Salicornia europaea) and perennial (Sarcocornia fruticosa, Arthrocaulon macrostachyum) glassworts collected and/or cultivated across multiple sites in the Apulia region (Italy). The study is timely and relevant, given the increasing need for resilient crops under salinity and water scarcity. The work is particularly strong in (i) covering both wild and cultivated systems; (ii) profiling a broad panel of traits (minerals, iodine, pigments, phenolics, flavonoids, sterols, and antioxidant capacity); and (iii) integrating univariate statistics with multivariate PCA. The clear evidence that genotype is the primary driver while being markedly modulated by site conditions is valuable for agronomy and product development. The dataset can support future guidelines on genotype and provenance selection for nutritional quality and food safety. |
Thank you for the positive comments. |
2 |
Abstract: When stating that Na/K “ exceeded the recommended ratio of 1, ” briefly contextualize by citing the health relevance (hypertension risk mitigation), and explicitly highlight the lowest ratio achieved in non-saline S. europaea as a practical target. |
We thank the reviewer for this valuable suggestion to improve the clarity and impact of our abstract. We have now revised the statement to contextualize the health relevance of the Na/K ratio concerning hypertension risk. Furthermore, we have explicitly highlighted the lowest ratio achieved in non-saline S. europaea and in an A. macrostachyum as a practical nutritional target, as recommended (lines 24-27). |
3 |
Introduction: Since iodine emerges as a key nutritional trait in your conclusions, add a sentence in the Introduction motivating why iodine in halophytes is of interest (human RDI, biofortification angle, and variability by provenance).; |
We thank the reviewer for this suggestion. To meet the comment, we have included a sentence. Please see lines 79-85… |
4 |
Introduction: You discuss antinutritional factors later (oxalate, Na/K). It would help to preview these early as part of the risk–benefit framework (e.g., importance of Na/K ratio and oxalate to human health). |
We thank the reviewer for this suggestion; we have modified the introduction including “However, a complete nutritional evaluation requires a risk-benefit analysis, which must also consider the presence of antinutritional factors such as oxalate, which can inhibit mineral absorption [19], and an elevated sodium-to-potassium (Na/K) ratio, a factor linked to hypertension risk [20].”. Please see lines 85-88. |
5 |
2.1: Seasonal/temporal confounding. Wild samples were collected April–June, while cultivated samples were collected August–September. Please discuss the potential confounding due to season on pigments/phenolics and, if possible, include this as a limitation or provide partial correction (e.g., analyzing a subset across overlapping months or citing literature on seasonal variation for these species). |
We thank the reviewer for this insightful comment. We agree that seasonal and interannual variability are crucial factors influencing metabolite profiles in halophytes. We have reported in the discussion section “In S. europaea samples, a further possible explanation for the higher TP levels could be related to the harvest period. Both cultivated glassworts were collected during the hottest months (August-September), while the wild samples were collected during the spring period. This seasonal variation may have influenced the plants' secondary metabolite production, including phenolics”. At line 714-718 in the text.
Further, we have explicitly acknowledged this limitation in the conclusion section of the manuscript (as shown in the new sentence below). We believe this addition provides important context for interpreting our results and suggests a valuable direction for future research. Added in the Conclusion section: “Although this study provides a snapshot of metabolite profiles during one spring–summer period, we recognize that these profiles may be subject to seasonal and interannual variation. Future studies involving multi-seasonal sampling would be valuable to elucidate the full dynamic range of metabolic responses in these halophyte species.”. Please see lines 881-884 |
6 |
3.1/Table 2: To facilitate health-related interpretation, consider adding a column for Na/K ratio and flagging samples closest to the target of 1. Also provide mean ± SD in addition to letters, and include LOD/LOQ footnotes where “0.0” or “n.d.” is reported. |
We thank the reviewer for this advice. We have modified the table 2 including the column Na/K as proposed. We also integrated the tables with the standard error of means, and the additional information in the table footnotes. We added a sentence in the result and discussion sections for Na/K parameter. Please see lines: 462-464, lines: 570-576, and the tables |
7 |
3.2/Table 4: Since hydrophilic antioxidants dominate, it would be informative to add correlations (Pearson/Spearman) between total phenols/flavonoids and HA-ABTS/FRAP across samples (include as a Supplementary figure/table). This will quantitatively support your discussion that phenolics likely drive HA responses. |
We thank the reviewer for this insightful comment. We have produced a supplementary table (Table S1), reporting Spearman coefficients as required. In the discussion section, the following has been added “In general, the correlation between the hydrophilic antioxidant capacity ranged from 0.61 to 0.76 for total phenols and from 0.59 to 0.84 for flavonoids (Table S1). The highest values detected for the DPPH test underlined a strong positive with phenols.” Please see lines 807-809 |
8 |
3.2: Sterols. Given the particularly high sterol value in A. macrostachyum-MSS, add a short paragraph discussing potential environmental drivers (aridity, temperature) and the relevance for functional foods (cholesterol-lowering claims), with a measured tone to avoid overstatement. |
We thank the reviewer for this insightful comment. We increased the discussions on sterols, adding: “Sterols, naturally present in plants, have anti-tumorigenic effects and prevent free radical damage by lowering blood cholesterol, resulting in reduced coronary heart disease [85]. A lowering effect of low-density lipoprotein cholesterol level in blood is reported for phytosterols [86]. In plants, they are essential in embryogenesis processes and for cellular membrane fluidity and permeability [87]. The biosynthesis of these compounds is influenced by both soil and climatic factors. For instance, Pavlík et al. [88] observed increases in sterol content in maize plants with improved nitrogen availability. Roche et al. [87,89] also provided evidence that environ-mental conditions like higher temperature and lower rainfall positively affected phy-tosterol content in sunflowers and safflowers. This underscores the role of phytosterols in maintaining proton balance within cells, regulating their activity, particularly in terms of membrane fluidity and permeability, helping the cells adapt to oxidative processes occur-ring in unfavorable environmental conditions [87]. Research on sterol changes in halophytes is limited. Samples of Salicornia perennis collected from the coastal lagoon of Rio Aveiro (Portugal) exhibited sterol concentrations of 210 mg g⁻¹ DW in sites characterized by lower salinity and more frequent flooding. However, in areas with elevated salinity and drier conditions, these concentrations soared to 600 mg g⁻¹ DW. It was suggested that the observed increase in sterols helped protect cell membrane lipids, thereby enhancing membrane resistance [90]. Also in Spartina patens, Rozentsvet et al. [91] reported that an increment in plasmalemma sterols was connected to rising NaCl concentrations in the growth medium. These studies highlight the role of sterols in maintaining membrane integrity during salt stress, even at the cost of increased membrane permeability. As reported for the other compounds mentioned so far, sterol content is significantly influenced by plant location, with the A. macrostachyum-MSS sample significantly exceed-ing other glassworts. This finding may be attributed to the characteristics of its sampling area, which is marked by a lack of water during certain times of the year. The presence of this species on a poorly watered site validates its capability to thrive in arid conditions. It should be noted that in the case of S. fruticosa collected from the same area (MSS), the exact spot was near the stream mouth, in a flooded area.” and “The high levels present in this population suggest A. macrostachyum could be a promising candidate for further investigation as a source of natural phytosterols. However, it is cru-cial to note that this study quantifies content rather than bio-efficacy; thus, while the po-tential for cholesterol-lowering claims exists, specific health benefits would require vali-dation through clinical studies.”. Please see lines 733-762 and 782-786 |
9 |
4.1 Mineral nutrients and antinutrients: The higher NO3− in S. europaea-SC and A. macrostachyum-MA-BC likely reflects fertilization/runoff. Consider adding a short note on how agronomic practices could be optimized to minimize nitrate accumulation while maintaining yield/quality.; |
Thank you for the comment. Despite the nitrate level in two glasswort samples being higher than others, we have clarified that “samples were well below European Community (EU 2023/915) safety thresholds for leafy vegetables”. The maximum value was close to 400 mg kg-1 fw, which is much lower than levels occurring in other leafy vegetables. It seems that these species are not prone to accumulating nitrate For clarity, we have rephrased the text as the following: “Our samples had markedly lower values of these thresholds; at least 400 mg NO3 kg-1 FW both in S. europaea-SC and A macrostachyum-MA-BC. This suggests that these species are not prone to accumulating nitrate”. Please see lines 604-607 |
10 |
Where you discuss Na/K > 1 and oxalate, consider adding approximate serving-size guidance (e.g., 50–100 g portions) and a brief note that culinary processing (blanching) can modulate Na and oxalate in some leafy vegetables, with appropriate references. |
Thank you for this insightful and valuable comment. Our study was primarily designed as a foundational biochemical characterization of Apulian glasswort, which, as you note, had not been previously documented. Within this scope, we focused on establishing the presence of key compounds and their potential implications for nutritional intake. Your points regarding serving-size guidance and culinary processing are excellent and highlight crucial next steps for translating these findings into practical dietary advice. We agree that these are critical areas for future research to build directly upon this initial characterization. |

Reviewer 3 Report
Comments and Suggestions for Authors
This study describes the nutritional characterization of perennial glassworts from the Apulia region, analyzing their mineral content, various antioxidative compounds, phytosterols, and potential antioxidative capacity using multiple assays. The manuscript is well-organized and well-written, with no major concerns. However, given the exceptionally high iodine and mineral levels in these perennials, it is recommended that the authors include several sentences discussing potential food applications, especially in the context of alternative protein-based products. Could the high iodine level and natural salty flavor be leveraged in the formulation of seafood analogues? This would give readers another perspective and exciting potential applications of these valuable plants. In addition, authors are encouraged to propose practical strategies for reducing antinutrient levels while preserving beneficial compounds, offering more actionable insights.
Author Response
|
Reviewer 3 |
|
1 |
This study describes the nutritional characterization of perennial glassworts from the Apulia region, analyzing their mineral content, various antioxidative compounds, phytosterols, and potential antioxidative capacity using multiple assays. The manuscript is well-organized and well-written, with no major concerns. |
Thank you for the positive comments |
2 |
However, given the exceptionally high iodine and mineral levels in these perennials, it is recommended that the authors include several sentences discussing potential food applications, especially in the context of alternative protein-based products. Could the high iodine level and natural salty flavor be leveraged in the formulation of seafood analogues? This would give readers another perspective and exciting potential applications of these valuable plants. |
The primary focus of our work is the biochemical characterization of glasswort samples from the Apulian region, highlighting the nutritional potential and any anti-nutrients in these plants, which has never been done before. Your points regarding potential food applications and the process analysis for alternative products are excellent and highlight crucial next steps for translating these findings into possible applications. We agree that these are critical areas for future research to build directly upon this initial characterization. |
3 |
In addition, authors are encouraged to propose practical strategies for reducing antinutrient levels while preserving beneficial compounds, offering more actionable insights. |
Thank the reviewer for posing this question; however, the primary focus of this work is the biochemical characterization of glasswort samples from the Apulian region. By the way, we have discussed some indications on reducing the main anti-nutrients found in our samples (Na, Cl, and oxalates). Specifically, from lines 576 to 578 for Na, from 585 to 587 for Cl, and from 613 to 616 for oxalates. |

Round 2
Reviewer 2 Report
Comments and Suggestions for Authors
yes